# Effect of Sleep Quality and Depression on Married Female Nurses’ Work–Family Conflict

**DOI:** 10.3390/ijerph18157838

**Published:** 2021-07-23

**Authors:** Eunhee Hwang, Yeongbin Yu

**Affiliations:** Department of Nursing, Wonkwang University, Iksan 54538, Korea; ehh@wku.ac.kr

**Keywords:** work, family, role conflict, depression, sleep quality, married nurses

## Abstract

Married female nurses experience work–family conflict (WFC) as they manage excessive work and various working-hour types while rearing children and tending household chores, and as a result, they continuously constantly deliberate over quitting their job or moving to a different workplace. Married nurses were found to have shorter sleep duration and sleep latency compared to single nurses, and high job stress not only hinders their family life but also causes sleep problems. Depression is a classic negative emotion experienced by married working women who must manage both work and family. This study aims to examine WFC in married female nurses and investigate its predictors, namely depression and sleep quality. A total of 229 married female nurses completed a Google questionnaire link consisting of the Work–Family Conflict Scale, Sleep-Quality Scale, and the Center for Epidemiologic Studies Depression Scale (CES-D). Data were analyzed by descriptive statistics, *t*-test, ANOVA, LSD post hoc test, Pearson’s correlation coefficients, and multiple regression using the SPSS/WIN 26.0 program. The average WFC score was 4.84 ± 1.12 (range 1–7); WFC showed a statistical difference according to a stage of the lifecycle (F = 7.12, *p* = 0.001) and perceived health (F = 12.01, *p* < 0.001). WFC was low among those in the non-parenthood stage of the lifecycle (β = −0.26, *p* < 0.001), those with good (β = −0.18, *p* = 0.011) or moderate perceived health (β = −0.15, *p* = 0.023), and those without turnover intention (β = −0.13, *p* = 0.016). On the other hand, WFC was high among those who were extremely dissatisfied with their job (β = 0.16, *p* = 0.008) and those who had a high level of depression (β = 0.22, *p* = 0.002); these variables explained 20.2% of WFC (F = 7.663, *p* < 0.001). Based on these results, subsequent studies should develop and implement coping programs that help reduce WFC and improve depression and sleep quality in married female nurses.

## 1. Introduction

The number of active nurses per 1000 people in the Republic of Korea (ROK) is low, at about 3.5, compared to other OECD countries, and the number of active nurses per bed is merely 18.6% of the OECD average, showing that nurses in the ROK are subject to high-intensity labor. In terms of the gender ratio of nurses in Korean hospitals, 95% or more are women, and 52.6% have less than 3–5 years of clinical experience. The average turnover rate of Korean nurses is 15.4%; their total clinical experience averages 7 years and 5 months, and they leave work by approximately 30 years of age [1,2]. In a meta-analysis of the factors related to the turnover intention of hospital nurses, the facilitating factors that increase their turnover intention in order of significance were emotional labor, role conflict, and work–family conflict. As for the reasons for turnover, “moving to another hospital” was the most common response at 16.7%, and “marriage, childbirth, and childcare” accounted for 15.9% [3]. Considering that the average age of first marriage among Korean women is 30.8 years [4], married female nurses form a family around this point of their lifecycle, and experience role conflicts at work and home, which in turn causes turnover intention. In terms of medical personnel, since married female nurses are crucial resources for high-quality medical services with skilled careers, it is essential to identify their difficulties, prevent turnover, and find ways to secure their continued employment in practice.

Married female nurses experience work–family conflict (WFC) as they manage excessive work and shift work while rearing children and tending household chores, and as a result they continuously and constantly deliberate over quitting their job or moving to a different workplace [5,6,7,8]. WFC is related to the gap in one’s roles and responsibilities between work and home, and it is a bidirectional role conflict in which family can interfere with work and work can interfere with family [5,9,10]. According to a previous study, 79.4% of nurses experienced WFC [11], and the prevalence of WFC was higher among female than male nurses, and among those with young children [12], showing that married female nurses are a highly vulnerable group for WFC. As WFC can lead to job stress, burnout, diminished work motivation, reduced performance, lower work satisfaction, absenteeism, and high turnover [5,7,10,12], WFC is a timely topic of discussion.

Married nurses were found to have shorter sleep duration and sleep latency compared to single nurses, and high job stress not only hinders their family life, but also causes sleep problems [13,14,15,16], highlighting the fact that married female nurses are more vulnerable to sleep problems than other groups. Diminished sleep quality decreases concentration, induces physical diseases, reduces work performance, and increases the risk of medical malpractice, such as medication error, error in medical-device manipulation, error in judgment of patients’ states, and needle injuries, as well as deteriorating the quality of care, such as displaying emotional inattention to patients, all of which have an adverse impact on the recovery of patients who received the care [16,17,18,19,20]. Therefore, nurses’ sleep quality is an important factor not only for their and patients’ health, but also for preventing medical malpractice. As WFC and parenting are negatively correlated with nurses’ sleep duration and quality, and WFC and poor sleep harm women’s health [14,16], it is necessary to comprehensively examine married female nurses’ sleep quality and its interaction with WFC.

Depression is the most common psychological problem suffered by nurses [21,22], and the incidence of depression is about twofold higher among nurses than in other professions due to the higher stress level [23,24]. The prevalence of depression is 18% among American nurses [22], 32% among Australian nurses [25], 38% among Chinese nurses [26], and 37.6–49.25% among Korean nurses [24], shedding light on the markedly higher rate of depression among Korean nurses. Depression brings about tremendous loss for the organization, as it has a negative impact on work performance, job satisfaction, and provision of nursing care for patients, and severe depression increases turnover intention by 4.60 times [20,22,24].

Further, nurses suffering from depression also experience physical symptoms such as fatigue, anxiety, headache, sudden cardiac arrest, and chronic diseases [22,23]. Depression is a classic negative emotion experienced by married working women who must manage both work and family, and the prevalence of depression is especially higher among married working women in the ROK compared to their counterparts in other countries [27]. This is because traditionally, married women in the ROK have played a pivotal role in housework and parenting, and even today, where female employment is considered a norm, working women still suffer from role conflicts between their traditional gender roles [27,28,29]. Depression diminishes marital satisfaction [6] and affects children among those who have children by leading to inappropriate parenting, such as inconsistent parenting style, punishment, abuse, and oppressive control [28,29]. Thus, depression in married female nurses should be dealt with priority in that it affects both the individuals as well as having an adverse impact on and conflict between their work and family.

Most married female nurses who manage both work and family are experienced nurses and key healthcare personnel with proficiency in nursing work, so efficient management of these nurses determines the quality of health service. Hence, helping married female nurses to carry on a healthy life by promoting successful management of family and work is an important factor not only for their individual lives, but also for their families, hospital organizations, and overall national competitiveness.

Thus, this study aims to examine WFC in married female nurses and investigate its predictors, namely depression and sleep quality, so as to provide empirical resources for improving their lives and quality of healthcare service by effectively managing WFC.

## 2. Materials and Methods

### 2.1. Design

The study population consisted of 300 married female nurses from general hospitals in three regions of Korea. The inclusion criteria were: married female nurses working at general hospitals in the three regions; married female nurses who live with their husbands; and nurses who had a minimum of 12 months of clinical career work experience.

This study was a descriptive online survey aiming to investigate the effects of sleep quality and depression on WFC in married female nurses who work in a general hospital.

### 2.2. Participants

Married female nurses with at least 1 year of clinical career at a general hospital in three regions were enrolled in this study. The sample size was determined using the G*Power 3.1.9.2 software. For multiple regression at a significance of 0.05, medium effect size of 0.15, power of 0.95, and with 23 predictor variables, the required sample size was 234. In consideration of a 25% potential dropout due to the nature of an online questionnaire, the Google questionnaire link was distributed to 300 married female nurses. A total of 299 questionnaires were retrieved, and after excluding nine questionnaires with incomplete responses, 290 questionnaires were included in the analysis.

### 2.3. Measurements

#### 2.3.1. Work–Family Conflict

WFC was measured using the Work–Family Conflict Scale developed by Netemeyer, Boles, and McMurrian [9] and adapted and modified by Choi [10] through a translation-back translation process. The modified tool contained five items for work–family conflict, and each item was rated on a 7-point Likert scale. A higher score indicated a greater work–family conflict. The Cronbach’s α was 0.94 at the time of development and 0.91 in this study.

#### 2.3.2. Sleep Quality

Sleep quality was measured using the tool developed by Oh et al. [30]. This 15-item tool uses a 4-point Likert scale from 1 (“strongly agree”) to 4 (“strongly disagree”), and negatively worded items were reverse-coded. The total score ranges from 15–60, and a higher score indicates better sleep quality. The Cronbach’s α was 0.75 at the time of development and 0.91 in this study.

#### 2.3.3. Depression

Depression was measured using the Center for Epidemiologic Studies Depression Scale (CES-D) developed by Radloff [31] and adapted by Cho and Kim [32]. This 20-item tool rates the frequency of depression symptoms experienced in the past week using a 4-point Likert scale, from “strongly disagree” to “strongly agree.” The total score ranges from 0–60, and a higher score indicates more severe depression. A score of 16 or higher indicates mild depression, and a score of 25 or higher indicates a clinical depression requiring pharmacological therapy and professional counseling. The Cronbach’s α was 0.91 in the study by Cho and Kim [32] and 0.91 in this study.

### 2.4. Data Collection

After approval by the Institutional Review Board (IRB), data were collected during December 2020. The researchers explained the purpose and method of the study to the managers of the general hospitals over the phone or in person, and the participants were recruited through the managers of three general hospitals that consented to participate in the study. The first page of the Google questionnaire was a consent form for the collection of personal information and confidentiality to ensure voluntary participation. The questionnaire responses were automatically processed by computer software such that personal identification was impossible.

### 2.5. Data Analysis

The collected data were analyzed using the SPSS/WIN 26.0 software. Participants’ general and work-related characteristics, sleep quality, depression, and WFC were analyzed with descriptive statistics, namely frequency with percentage and mean with standard deviation. Differences in WFC according to general characteristics were analyzed with *t*-test, ANOVA, and LSD post hoc test. The correlations among sleep quality, depression, and WFC were analyzed with Pearson’s correlation coefficients, and the factors that affected WFC were analyzed with multiple regression.

## 3. Results

### 3.1. General Characteristics

The mean age was 37.78 (±8.39) years, and the majority of the participants had a bachelor’s degree (*n* = 214, 73.8%). A total of 255 (87.9%) were in a double-income family, and the most common monthly household income was ≥ KRW 6 million (*n* = 175, 60.3%), followed by KRW 5–5.9 million (*n* = 52, 17.9%) and KRW 4–4.9 million (*n* = 38, 13.1%). Stage of family’s lifecycle was school-aged or older child (*n* = 122, 42.1%), young child (*n* = 114, 39.3%), or non-parenthood (*n* = 54, 18.6%), and the majority of the participants had two children (*n* = 161, 67.6%). Regarding satisfaction with spouse’s household and parenting involvement, 108 (37.2%) were satisfied, 83 (28.6%) were neutral, and 41 (14.1%) were dissatisfied. A total of 137 (47.2%) participants perceived themselves to be in moderate health, while 87 (30.0%) perceived themselves to be in poor health. The most common sleep duration was 6–6.9 h (*n* = 94, 32.4%). A total of 196 (67.6%) participants had irregular meal patterns, and 50 (17.2%) engaged in regular exercise (Table 1).

### 3.2. Participants’ Work-Related Characteristics

The most common hospital size was 800 beds or more (*n* = 145, 50.0%), followed by 400–799 beds (*n* = 128, 44.1%) and 100–399 beds (*n* = 17, 5.9%). Most of the participants were staff nurses (*n* = 224, 77.2%), and regarding job type, 118 (40.7%) worked regular office hours, while 172 (59.3%) worked rotating shifts. A total of (45.2%) participants worked night shifts, 168 (57.9%) currently worked in a general ward, and 84 (29.0%) worked in other units (recovery room, operation room, delivery room, or neonatal room). Regarding satisfaction with their current job, 155 (53.4%) said neutral, followed by 82 (28.3%) satisfied, 37 (12.8%) dissatisfied, 10 (3.4%) extremely dissatisfied, and 6 (2.1%) extremely satisfied. When asked about whether they were considering leaving the job, 251 (86.6%) said yes. The mean total clinical career was 202.55 (±121.59) months, and the mean clinical career at the current unit was 58.91 (±77.73) months. The monthly average number of night shifts was 5.44 (±1.72) (Table 2).

### 3.3. Level of WFC, Sleep Quality, and Depression

The mean WFC score was 4.84 (±1.12) out of 7, and the mean sleep-quality score was 37.45 (±7.87) out of 60. The mean depression score was 35.46 (±9.63) out of 60. Based on the depression score, 26 (9.0%) had mild depression, while 264 (91.0%) had severe depression (Table 3).

### 3.4. Differences in WFC According to General Characteristics

Participants’ WFC significantly differed according to a stage of the family lifecycle (F = 7.12, *p* = 0.001) and perceived health (F = 12.01, *p* < 0.001) (Table 4). The post hoc test confirmed that participants with a young child or school-aged or older child had greater WFC compared to participants in non-parenthood. Further, participants with poor perceived health had significantly greater WFC compared to those who had moderate or good perceived health.

### 3.5. Differences in WFC According to Work-Related Characteristics

Participants’ WFC significantly differed according to satisfaction with their current job (F = 7.95, *p* < 0.001) and turnover intention (t = 2.52, *p* = 0.012) (Table 5). The post hoc test confirmed that participants who were extremely dissatisfied with their current job suffered from greater WFC compared to those who claimed otherwise. Further, participants who were considering turnover had greater WFC compared to those who are not.

### 3.6. Correlations among WFC, Sleep Quality, and Depression

WFC was significantly negatively correlated with sleep quality (r = −0.14, *p* = 0.018) and significantly positively correlated with depression (r = 0.34, *p* < 0.001). Further, sleep quality was negatively correlated with depression (r = −0.52, *p* < 0.001) (Table 6).

### 3.7. Factors Affecting WFC

To identify the factors that affected WFC, a regression analysis was performed with a stage of the lifecycle, perceived health, satisfaction with current job, turnover intention, sleep quality, and depression, all of which significantly differed in relation to WFC as the independent variables, and with WFC as the dependent variable (Table 7). The stage of lifecycle, perceived health, satisfaction with current job, and turnover intention were dummy-coded. The Durbin–Watson statistic, which represents the autocorrelation of errors of the variables included in the model, was close to 2, confirming the absence of autocorrelation. The variance inflation factor (VIF) was below 10, confirming the absence of multicollinearity. The F statistic of the regression model was significant at 7.663, *p* < 0.001, and the coefficient of determination (R^2^) was 0.202, suggesting that the model explained 20.2% of the variance of WFC. WFC was low among those in the non-parenthood stage of the lifecycle (β = −0.26, *p* < 0.001), those with good (β = −0.18, *p* = 0.011) or moderate perceived health (β = −0.15, *p* = 0.023), and those without turnover intention (β = −0.13, *p* = 0.016). On the other hand, WFC was high among those who were extremely dissatisfied with their job (β = 0.16, *p* = 0.008) and those who had a high level of depression (β = 0.22, *p* = 0.002).

## 4. Discussion

This was a descriptive survey aiming to investigate the effects of sleep quality and depression on WFC in married female nurses who work in a general hospital.

The mean age of participants was 37.78 years. A total of 87.9% of the participants were in a double-income family, and 81.4% had children. Despite the fact that 16.2% of the participants claimed to be dissatisfied with their current job, a whopping 86.6% of the participants had a turnover intention, calling for further research to identify factors other than job satisfaction that influence married female nurses’ turnover intention. This rate of turnover intention was high compared to foreign countries, as the rate of turnover intention was 59.9% among nurses of long-term care hospitals in Japan [33], 20.9% among American nurses in consideration of an appropriate level of nursing staffing, and 45.6% in consideration of an inappropriate level of nursing staffing [34]. High turnover intention is likely to lead to turnover, which results in a shortage of experienced nurses in the hospital and an increased patient-to-nurse ratio, leading to adverse outcomes such as deteriorated quality of care. Hence, it is necessary to identify the causes of their turnover intention and develop interventions for them.

The mean WFC score among participants was 4.84 out of 7, indicating a level of WFC above the midline. This score was higher than the score of 4.4 reported by Lee and Ko [35] among married nurses using the same instrument, which was presumably due to the differences in the study population, as the said study included male nurses as well. Further research is needed to examine gender-specific differences. In foreign countries, the WFC score was reported to be 2.49 among American nurses [36], 3.3 among Japanese nurses [8], and 3.21 among Taiwanese nurses [37], showing that married female nurses in the ROK experience greater WFC than their counterparts in other countries. This may be attributable to cultural differences in the traditional gender role, as Korean women have traditionally performed central roles in housework and parenting, and they are faced with role conflicts even today, when female employment has become a norm [6,28]. The differences in WFC across countries may be due to cultural diversity pertaining to women’s social position or role conflict, and replication studies are needed to substantiate this.

The mean sleep-quality score in this study was 37.45 out of 60. This was higher than 31.83 among shift-working nurses [12] and 36.6 among shift-working nurses in university hospitals, and lower than 43.5 among non-shift-workers in university hospitals [19]. Considering that 45.2% of our participants were shift workers, we can speculate on the sleep quality of married female nurses. It is difficult for married women to have quality sleep due to household chores and parenting. Such deterioration of sleep quality in married female nurses is directly linked to patient safety as well as nurses’ own health, and because poorer sleep quality is associated with higher turnover intention [13] and has socioeconomic repercussions, individuals’ efforts, as well as interventions that improve sleep quality, are also needed in terms of work–family balance.

In our study, 91.0% of the participants had severe depression. This was a higher rate than that reported among shift-working nurses using the same tool (71.2%) [35] and that reported among comprehensive nursing care service nurses (63.6%) [38]. This difference seems to be due to the fact that our participants only comprised married female nurses in general hospitals, and so it was necessary to identify the factors that induced depression in married female nurses and develop tailored nursing intervention strategies for this group.

Our participants experienced greater WFC with poorer sleep quality and more severe depression. This correlation was consistent with previous findings that WFC was positively correlated with sleep problems [14,16]. Married female nurses experienced changes in their sleep patterns and poorer sleep quality due to shift work or strenuous work, which in turn affected their daily living and family, thereby intensifying WFC [13,16,19,20,36]. Furthermore, this induced memory and concentration loss and increased tension, which reduces work efficiency, such as medication errors, diminished performance, and errors in patient identification [15,16,17,18,19,20]. Moreover, poor sleep quality has an adverse impact on physical health, such as causing coronary artery disease or gastrointestinal disorders [13], calling for aggressive management and intervention. Sleep problems serve as barriers to achieving a good work–family balance and promoting good health in married female nurses. Thus, to improve sleep quality, family support is crucial, and interventions that involve family members regarding the management of sleep problems should be considered.

Married female nurses’ sleep quality exacerbated WFC and diminished their work efficiency, and thus had a strong correlation with depression, but sleep quality was not a significant predictor of WFC. This may be attributable to the fact that 54.8% of our participants worked regular office hours without night shifts, and because we could not adequately examine an array of factors that may influence WFC. Hence, further research is needed to examine sleep quality using various instruments.

In this study, lifestyle was the most potent predictor of WFC, where nurses in the non-parenthood stage of lifecycle had lower WFC. This was consistent with the findings of a study on married workers, in which those with many children or young children were more vulnerable to WFC, as they needed to spend a lot of time in parenting and family care [39]. In a study of married female physicians with children, “subjective reduced-hours job-role quality” was an important variable in predicting life satisfaction, supporting the results of this study [40].

As nurses are predominantly female, the responsibility of parenting cannot be considered an individual or family problem in order to reduce their stress related to childbirth and parenting and promote a good work–family balance in married female nurses. Practical and effective institutional policies that allow nurses to participate in family life and reduce WFC, such as parental leave and flexible work schedules, are needed. In particular, prior research has shown that working schedules are the most important factor for preventive health behavior, and the subjective health status of married working women with young children supports this [41].

In this study, WFC was lower among those with moderate or good perceived health than among those with poor perceived health. This was similar to previous results showing that the odds for poor perceived health are 2.8 times higher among married working women with a high level of WFC [39]. In a study on nurses in Italy, WFC was found to be significantly associated with somatization [21], and a study on workers of a general hospital reported that perceived health affects health-promoting behaviors, and that a health-promoting behavior program increased nurses’ compliance with health-promoting behaviors [42]. Thus, including health-promoting education in WFC intervention programs for married female nurses could be more effective, along with perceived health status, in reducing WFC.

A previous study reported that WFC increased with decreasing job satisfaction, and that job satisfaction played a key role in WFC and had a grave impact on nurses’ physical and psychological health [5,43], which was consistent with our results that job satisfaction was a predictor of WFC. Managing job satisfaction is crucial, as it lowers turnover and enhances the quality of care [43,44]. In our study, nurses without turnover intention displayed a low level of WFC, and this was similar to past findings that high WFC led to increased job dissatisfaction and stress, and thus was strongly correlated with turnover intention [5,39,43,45]. Low turnover intention among married female nurses can be understood as increased job satisfaction and work motivation and relatively low stress, suggesting that this reduces their role conflict between work and family.

In terms of the psychological aspect of married female nurses, greater levels of depression affected WFC, and there was a significant positive correlation between depression and WFC. These results were consistent with previous findings that depression symptoms were associated with WFC [28,36]. The traditional gender role for women in the ROK is to impeccably manage family-related matters, even if they work. Hence, married women in the ROK suffer from WFC due to their compulsion that they must be perfect in both their work and home, and frustration of such desire readily triggers negative emotions such as depression. The results of a study that compared South Korea, the US, and Israel, in which the working mothers in the ROK had the greatest level of depression, supported this [27]. Specifically, a previous study reported that female healthcare workers had a higher suicide risk than male healthcare workers. Additionally, the suicide rate of female healthcare workers was twice as high as that of women in the general population, suggesting that married female nurses had more severe levels of depression [46]. According to a longitudinal study of depressive symptoms and WFC in married women, the most crucial predictor of WFC was family-related rather than work-related [47]. This was consistent with the results of this study, showing that the family lifecycle was the most important influencing factor. Therefore, a family-friendly policy is necessary to achieve a work–family balance that positively affects the attitudes and behaviors of workers and increases organizational commitment, job satisfaction, and life satisfaction.

There were some limitations to this study. First, this study used a self-report questionnaire, and as such, self-selection bias could not be ruled out. Second, the participants in this study were only Koreans; thus, there was a limit to generalizing these groups globally, owing to Korea’s unique cultural characteristics. Therefore, further studies should be conducted in different cultural contexts. Third, this study aimed to identify factors affecting WFC, and we could not examine the differences in depression and sleep quality according to the type of work or shift (night vs. day). Future research is needed to confirm the effect of work or shift type on depression and sleep quality in married female nurses. Fourth, although the relationship with various variables was considered in this study, the explanatory power of the variables affecting the WFC was only an average of 20%. Therefore, further research with a larger sample of married female nurses is needed. Additionally, it is necessary to include social resource-related variables and other variables that potentially affect the WFC of married female nurses.

Despite these limitations, this study was significant in that it identified family conflicts among married female nurses and the factors affecting them. This has practical implications for the work–family reconciliation of married female nurses. Managers should provide emotional care for married female nurses by expressing that they care about individuals and their families, and their desire to balance work and family roles. Supporting flexible and diverse work types and childcare policies, reducing turnover intention, and increasing job satisfaction can improve work performance and quality of care.

## 5. Conclusions

This study found that WFC was intensified with poorer sleep quality and higher severity of depression among married female nurses. Further, parenthood stage of the lifecycle, perceived health, job satisfaction, turnover intention, and depression were identified as the predictors of WFC in married female nurses, with lifecycle being the most potent predictor. In other words, the greatest stressor for married female nurses was having to do household chores and care for children after work, and thus married female nurses’ parenting stress needs attention in terms of their WFC. To address this issue, flexible work hours, a considerate work culture, on-site childcare services, and institutional support for parental leave are needed.

Based on these results, we present the following suggestions. First, our use of a subjective tool for measuring sleep quality diminished the accuracy of the data, so comparative studies should be conducted using objective sleep tools. Second, subsequent studies should also examine the factors related to WFC in married male nurses. Third, subsequent studies should develop and implement coping programs that help reduce WFC and improve depression and sleep quality in married female nurses.

## Figures and Tables

**Table 1 ijerph-18-07838-t001:** Participants’ general characteristics (*n* = 290).

Characters	Categories	Range	*n* (%), Mean ± SD
Age (years)		24–58	37.78 ± 8.39
Education level	Associate’s		40 (13.8)
	Bachelor’s		214 (73.8)
	≥Master’s		36 (12.4)
Income type	Double		255 (87.9)
	Single		35 (12.1)
Monthly household income (Million KRW)	<4		25 (8.7)
	4–4.9		38 (13.1)
	5–5.9		52 (17.9)
	≥6		175 (60.3)
Stage of family’s lifecycle	Non-parenthood		54 (18.6)
	Young child		114 (39.3)
	School aged or older child		122 (42.1)
Number of children (person) (*n* = 238)	1		51 (21.4)
	2		161 (67.6)
	≥3		26 (11.0)
Satisfaction with spouse’s household and parenting involvement	Extremely satisfied		29 (10.0)
	Satisfied		108 (37.2)
	Neutral		83 (28.6)
	Dissatisfied		41 (14.1)
	Extremely dissatisfied		15 (5.2)
	Missing		14 (4.8)
Perceived health status	Good		66 (22.8)
	Moderate		137 (47.2)
	Poor		87 (30.0)
Average sleep duration (hours)	<5		14 (4.8)
	5–5.9		8 (20.1)
	6–6.9		94 (32.4)
	7–7.9		74 (25.5)
	≥8		50 (17.2)
Meal pattern	Regular		94 (32.4)
	Irregular		196 (67.6)
Regular exercise	Yes		50 (17.2)
	No		240 (82.8)

**Table 2 ijerph-18-07838-t002:** Participants’ work-related characteristics (*n* = 290).

Characters	Categories	Range	*n* (%), Mean ± SD
Hospital size (beds)	100–399		17 (5.9)
	400–799		128 (44.1)
	≥800		145 (50.0)
Current position	Staff nurse		224 (77.2)
	Charge nurse		39 (13.4)
	≥Head nurse		27 (9.3)
Job type	Rotating shift		172 (59.3)
	Regular office hours		118 (40.7)
Night shift	Yes		131 (45.2)
	No		159 (54.8)
Working department	General ward		168 (57.9)
	Intensive care unit		24 (8.3)
	Emergency room		14 (4.8)
	Other units		84 (29.0)
Satisfaction with job	Extremely satisfied		6 (2.1)
	Satisfied		82 (28.3)
	Neutral		155 (53.4)
	Dissatisfied		37 (12.8)
	Extremely dissatisfied		10 (3.4)
Considering leaving the job	Yes		251 (86.6)
	No		39 (13.4)
Total clinical career (months)		12–1058	202.55 ± 121.59
Clinical career at the current unit (months)		2–569	58.91 ± 77.73
Monthly average number of night shifts		1–8	5.44 ± 1.72

**Table 3 ijerph-18-07838-t003:** Level of WFC, sleep quality, and depression (*n* = 290).

Characters	Min	Max	Mean ± SD	*n* (%)
WFC	1.00	7.00	4.84 ± 1.12	
Sleep quality	16.00	59.00	37.45 ± 7.87	
Depression	20.00	72.00	35.46 ± 9.63	
Mild				26 (9.0)
Severe				264 (91.0)

WFC, work–family conflict.

**Table 4 ijerph-18-07838-t004:** Differences in WFC according to general characteristics (*n* = 290).

Characters	Categories	Mean ± SD	t/F(*p*) LSD
Education level	Associate’s	35.08 ± 7.70	0.82 (0.441)
	Bachelor’s	37.43 ± 7.36	
	≥Master’s	40.17 ± 10.08	
Income type	Double	4.86 ± 1.12	0.73 (0.469)
	Single	4.71 ± 1.14	
Monthly household income (Million KRW)	<4	4.85 ± 1.22	0.26 (0.853)
	4–4.9	4.84 ± 1.14	
	5–5.9	4.96 ± 1.05	
	≥6	4.80 ± 1.13	
Stage of family’s lifecycle	Non-parenthood ^a^	4.33 ± 1.11	7.12 (0.001)
	Young child ^b^	4.96 ± 1.16	a < b,c
	School aged or older child ^c^	4.95 ± 1.03	
Number of children (person) (*n* = 238)	1	5.19 ± 0.93	1.47 (0.233)
	2	4.90 ± 1.11	
	≥3	4.88 ± 1.27	
Satisfaction with spouse’s household and parenting involvement	Extremely satisfied	4.72 ± 1.39	1.68 (0.156)
	Satisfied	4.84 ± 1.08	
	Neutral	4.85 ± 1.01	
	Dissatisfied	4.88 ± 1.09	
	Extremely dissatisfied	5.56 ± 0.96	
Perceived health status	Good ^a^	4.01 ± 1.24	12.01 (<0.001)
	Moderate ^b^	4.71 ± 0.99	c > a,b
	Poor ^c^	5.30 ± 1.08	
Average sleep duration (hours)	<5	5.25 ± 1.11	1.337 (0.256)
	5–5.9	4.95 ± 1.21	
	6–6.9	4.72 ± 1.14	
	7–7.9	4.72 ± 1.01	
	≥8	5.00 ± 1.12	
Meal pattern	Regular	4.68 ± 1.04	−1.65 (0.099)
	Irregular	4.91 ± 1.16	
Regular exercise	Yes	4.66 ± 1.06	−1.26 (0.209)
	No	4.88 ± 1.14	

WFC, work–family conflict.

**Table 5 ijerph-18-07838-t005:** Differences in WFC according to work-related characteristics (*n* = 290).

Characters	Categories	Mean ± SD	t/F(*p*) LSD
Hospital size (beds)	100–399	4.54 ± 0.99	0.67 (0.513)
	400–799	4.84 ± 1.12	
	≥800	4.84 ± 1.12	
Current position	Staff nurse	4.74 ± 1.15	1.62 (0.200)
	Charge nurse	5.13 ± 0.93	
	≥Head nurse	4.84 ± 1.08	
Job type	Rotating shift	4.87 ± 1.11	0.56 (0.575)
	Regular office hours	4.79 ± 1.14	
Night shift	Yes	4.84 ± 1.13	0.05 (0.964)
	No	4.84 ± 1.12	
Working department	General ward	4.92 ± 1.15	0.90 (0.444)
	Intensive care unit	4.91 ± 1.12	
	Emergency room	4.63 ± 0.97	
	Other units	4.70 ± 1.10	
Satisfaction with job	Extremely satisfied ^a^	4.77 ± 1.91	7.95 (<0.001)
	Satisfied ^b^	4.46 ± 1.12	c,d,e > b
	Neutral ^c^	4.85 ± 1.06	e > a,b,c,d
	Dissatisfied ^d^	5.24 ± 0.88	
	Extremely dissatisfied ^e^	6.22 ± 0.94	
Considering leaving the job	Yes	4.90 ± 1.09	2.52 (0.012)
	No	4.42 ± 1.28	

WFC, work–family conflict.

**Table 6 ijerph-18-07838-t006:** Correlations among WFC, sleep quality, and depression (*n* = 290).

Variables	Sleep Quality	Depression
	r(*p*)	
WFC	−0.14 (0.018)	0.34 (<0.001)
Sleep quality		−0.52 (<0.001)

WFC, work–family conflict.

**Table 7 ijerph-18-07838-t007:** Factors affecting WFC (*n* = 290).

Variables	B	SE	β	t	*p*
Variables	4.01	0.56		7.11	<0.001
Stage of family’s lifecycle = Non-parenthood	−0.75	0.18	−0.26	−4.23	<0.001
Perceived health status = Good	−0.49	0.19	−0.18	−2.55	0.011
Perceived health status = Moderate	−0.34	0.15	−0.15	−2.28	0.023
Satisfaction with job = Extremely dissatisfied	1.01	0.38	0.16	2.67	0.008
Considering leaving the job = No	−0.43	0.18	−0.13	−2.41	0.016
Depression	0.03	0.01	0.22	3.18	0.002
R^2^ = 0.233, Adj R^2^ = 0.202, F = 7.663, *p* < 0.001					

WFC, work–family conflict.

## Data Availability

The data presented in this study are available upon request from the corresponding author. The data are not publicly available due to participants’ privacy.

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
