# Peer review of "Effect of Sleep Quality and Depression on Married Female Nurses’ Work–Family Conflict"

_ijerph, 2021, doi:10.3390/ijerph18157838_

Round 1

Reviewer 1 Report

Thank you very much for giving me the opportunity to read this article. 
1. p. 6, table 2. working department: special unit - Please describe exactly which special department you are referring to other than ICU and ER.
2. Please indicate the range of age, total clinical career, and clinical career at the current unit.
3. The discussion section should describe the limitations of this study. In particular, despite including various independent variables, the explanatory power is about 20%. The limitations of this study, including these limitations, should be described in the discussion, and follow-up studies should be suggested.

Author Response

Dear Editor and Reviewers,

We would like to thank you for your thorough review of our manuscript and the opportunity to submit a revised and improved version of “Effect of sleep quality and depression on married female nurses’ work-family conflict”. We have provided point-by-point responses to the reviewers’ comments below. We have made the suggested changes in the manuscript and highlighted them for ease of review.  

1st Editor's comments:

Point 1. Table 2. working department: special unit - Please describe exactly which special department you are referring to other than ICU and ER.

Response 1: Thank you for your comment. We revised “special units” among the
sub-items of the work department to “other units,” and described the types of
other units to clarify the differences among them. The “other units” include
recovery room, operation room, delivery room, and neonatal room.

Page 5, Line 202 and Page 6, Table 2 and Page 8, Table 5

   A total of (45.2%) participants worked night shifts, 168 (57.9%) currently worked in a general ward, and 84 (29.0%) worked in other units (recovery room, operation room, delivery room, or neonatal room).

Response 2: Based on your advice, we have added the age range, total clinical career, and clinical career at the current unit (Tables 1 and 2).

Point 3. The discussion section should describe the limitations of this study. In particular, despite including various independent variables, the explanatory power is about 20%. The limitations of this study, including these limitations, should be described in the discussion, and follow-up studies should be suggested.

Response 3: We thank the reviewer for this comment. We agree with these
points. To reflect your comments, we have added limitations, including
limitations on explanatory power, to the Discussion (page 12).

Page 11, Lines 382-401.

     There were some limitations to this study. First, this study used a self-report questionnaire, as such, self-selection bias could not be ruled out. Second, the participants in this study were only Koreans. Thus, there is a limit to generalizing these groups globally due to the influence of Korea's unique cultural characteristics. Consequently, further studies should be conducted in different cultural contexts. Third, this study aimed to identify factors affecting WFC, and we could not examine the differences in depression and sleep quality according to the type of work or night shift. Future research is needed to confirm the effect of working type or night shift on depression and sleep quality in married female nurses. Fourth, although the relationship with various variables was considered in this study, the explanatory power of the variables affecting the WFC was only an average of 20%. Therefore, further studies with a larger sample of married female nurses is needed. Additionally, it is necessary to include social resource-related variables and other variables that potentially affect the WFC of married female nurses.

     Despite these limitations, this study is significant in that it identified family conflicts among married female nurses and the factors affecting them. This has practical implications for the work-family reconciliation of married female nurses. Managers should provide emotional care for married female nurses by expressing that they care about individuals and their families and their desire to balance work and family roles. Supporting flexible and diverse work types and childcare policies, reducing turnover intention, and increasing job satisfaction can improve work performance and quality of care.

Reviewer 2 Report

Thank you for the opportunity to read an interesting work. An interesting topic for me. The question is whether interesting for other readers. It is worth considering what is the source of depression among this professional group? In my opinion, it will be interesting to compare it with other people in a different industry or other groups in the medical industry. Additionally, it may be worth comparing night work with day work as a source of sleep problems or depression. Why married nurses? and whether it is worth comparing married women to unmarried women. The discussion largely repeats the results of the study. this part of the work needs improvement. 'The study lacked practical implications that would indicate the importance of the research conducted and the possibility of using the obtained results. Due to the correct methodology, selection of appropriate and interesting research tools, the work after making corrections can be considered for publication 

Author Response

We would like to thank you for your thorough review of our manuscript

and the opportunity to submit a revised and improved version of “Effect of sleep quality and depression on married female nurses’ work-family conflict”. We have provided point-by-point responses to the reviewers’ comments below. We have made the suggested changes in the manuscript and highlighted them for ease of review. 

2nd Editor's comments:

Point 1. Why married nurses?

Response 1: We thank the reviewer for this comment. As a result of referring to the current status of hospital nurses in Korea, turnover-related statistics, and previous studies on factors related to turnover intention, we judged that married female nurses are the most vulnerable group in terms of WFC. As the reviewer pointed out, the introduction and the research participant section of the research method were also revised to add more details (Pages 1–3).            

  • Pages 1~2, Lines 34-48:

             In terms of the gender ratio of nurses in Korean hospitals, 95% or more are women and 52.6% have less than 3–5 years of clinical experience. The average turnover rate of Korean nurses is 15.4%; their total clinical experience averages seven years and five months, and they leave work by approximately 30 years of age [1,2]. In a meta-analysis of the factors related to the turnover intention of hospital nurses, the facilitating factors that increase their turnover intention in order of significance were emotional labor, role conflict, and work-family conflict. As for the reasons for turnover, “moving to another hospital” was the most common response at 16.7%, and “marriage, childbirth, and childcare” accounted for 15.9% [3]. Considering that the average age of first marriage among Korean women is 30.8 years [4], married female nurses form a family around this point of their life cycle, experience role conflicts at work and home, which in turn causes turnover intention. In terms of medical personnel, since married female nurses are crucial resources for high-quality medical services with skilled careers, it is essential to identify their difficulties, prevent turnover, and find ways to secure their continued employment in practice.

  • Page 3, Lines 107-110:
  1. Materials and Methods

      2.1. Design

       The study population consisted of 300 married female nurses from general hospitals in three regions of Korea. The inclusion criteria were: married female nurses working at general hospitals in the three regions; married female nurses who live with their husbands; and, nurses who have a minimum of 12 months of clinical career work experience.

Point 2. Comparing night work with day work as a source of sleep problems or depression

Response 2: Thank you for your sincere comments. This study focused on family conflict, and depression and sleep quality were considered as influencing factors. Therefore, in this study, we did not analyze the difference in depression and sleep quality according to the type of work or night vs. day shift, but instead presented this as a limitation of the study and suggested that further research explore this aspect.

Point 3. Compare it with other people in a different industry or other groups in the medical industry, add practical implications discussion.

Response 3: We appreciate this comment. Based on your advice, in the discussion section, practical implications were described based on the results of the study through previous studies of other groups in the medical field and groups supporting the shift worker population (Pages, 10–12).

  • Page 10, Lines 332–334:

        In a study of married female physicians with children, “subjective reduced-hours job-role quality” was an important variable in predicting life satisfaction, supporting the results of this study [40].

  • Pages 10-11, Lines 339–342:

         In particular, prior research has shown that working schedules are the most important factor for preventive health behavior, and the subjective health status of married working women with young children supports this [41].

  • Pages 11-12, Lines 371–400

          Specifically, a previous study reported that female healthcare workers had a higher suicide risk than male healthcare workers. Additionally, the suicide rate of female health care workers was twice as high as that of women in the general population, suggesting that married female nurses had more severe levels of depression [46]. According to a longitudinal study of depressive symptoms and WFC in married women, the most crucial predictor of WFC was family related rather than work related [47]. This is consistent with the results of this study, showing that the family life cycle is the most important influencing factor. Therefore, a family-friendly policy is necessary to achieve a work-family balance that positively affects the attitudes and behaviors of workers and increases organizational commitment, job satisfaction, and life satisfaction. There were some limitations to this study. First, this study used a self-report questionnaire, as such, self-selection bias could not be ruled out. Second, the participants in this study were only Koreans; thus, there is a limit to generalizing these groups globally owing to Korea's unique cultural characteristics. Therefore, further studies should be conducted in different cultural contexts. Third, this study aimed to identify factors affecting WFC and we could not examine the differences in depression and sleep quality according to the type of work or shift (night vs. day). Future research is needed to confirm the effect of work or shift type on depression and sleep quality in married female nurses. Fourth, although the relationship with various variables was considered in this study, the explanatory power of the variables affecting the WFC was only an average of 20%. Therefore, further research with a larger sample of married female nurses are needed. Additionally, it is necessary to include social resource-related variables and other variables that potentially affect the WFC of married female nurses.

         Despite these limitations, this study is significant in that it identified family conflicts among married female nurses and the factors affecting them. This has practical implications for the work-family reconciliation of married female nurses. Managers should provide emotional care for married female nurses by expressing that they care about individuals and their families and their desire to balance work and family roles. Supporting flexible and diverse work types and childcare policies, reducing turnover intention, and increasing job satisfaction can improve work performance and quality of care.
